# Metagenomic Detection of Divergent Insect- and Bat-Associated Viruses in Plasma from Two African Individuals Enrolled in Blood-Borne Surveillance

**DOI:** 10.3390/v15041022

**Published:** 2023-04-21

**Authors:** Gregory S. Orf, Ana Olivo, Barbara Harris, Sonja L. Weiss, Asmeeta Achari, Guixia Yu, Scot Federman, Dora Mbanya, Linda James, Samuel Mampunza, Charles Y. Chiu, Mary A. Rodgers, Gavin A. Cloherty, Michael G. Berg

**Affiliations:** 1Infectious Disease Research, Abbott Diagnostics, Abbott Park, IL 60004, USA; 2Abbott Pandemic Defense Coalition, Abbott Park, IL 60004, USA; 3Department of Laboratory Medicine, University of California-San Francisco, San Francisco, CA 94143, USA; 4Faculty of Medicine and Biomedical Sciences, University of Yaoundé I, Yaoundé P.O. Box 1364, Cameroon; 5School of Medicine, Université Protestante au Congo, Kinshasa P.O. Box 4745, Democratic Republic of the Congo; 6Department of Medicine, University of California-San Francisco, San Francisco, CA 94143, USA

**Keywords:** bastrovirus, cyclovirus, dicistrovirus, metagenomic next-generation sequencing, viral surveillance, viral pathogen discovery

## Abstract

Metagenomic next-generation sequencing (mNGS) has enabled the high-throughput multiplexed identification of sequences from microbes of potential medical relevance. This approach has become indispensable for viral pathogen discovery and broad-based surveillance of emerging or re-emerging pathogens. From 2015 to 2019, plasma was collected from 9586 individuals in Cameroon and the Democratic Republic of the Congo enrolled in a combined hepatitis virus and retrovirus surveillance program. A subset (n = 726) of the patient specimens was analyzed by mNGS to identify viral co-infections. While co-infections from known blood-borne viruses were detected, divergent sequences from nine poorly characterized or previously uncharacterized viruses were also identified in two individuals. These were assigned to the following groups by genomic and phylogenetic analyses: densovirus, nodavirus, jingmenvirus, bastrovirus, dicistrovirus, picornavirus, and cyclovirus. Although of unclear pathogenicity, these viruses were found circulating at high enough concentrations in plasma for genomes to be assembled and were most closely related to those previously associated with bird or bat excrement. Phylogenetic analyses and in silico host predictions suggested that these are invertebrate viruses likely transmitted through feces containing consumed insects or through contaminated shellfish. This study highlights the power of metagenomics and in silico host prediction in characterizing novel viral infections in susceptible individuals, including those who are immunocompromised from hepatitis viruses and retroviruses, or potentially exposed to zoonotic viruses from animal reservoir species.

## 1. Introduction

The zoonotic transmission of novel viruses into the human population represents a substantial risk to public health [1,2]. Spillover events are most often observed in areas in which humans live in close proximity to reservoir species such as non-human primates, mosquitoes, bats, and rodents [3]. Well-known examples include the original emergence of human immunodeficiency virus (HIV), Ebola virus, and the severe acute respiratory syndrome coronaviruses (SARS-CoVs) [4,5,6,7]. Some zoonotic events rely upon intermediate hosts (e.g., West Nile virus, transmitted from birds to humans via mosquitoes [8]), some are sylvatic (e.g., dengue virus, transmitted between humans and/or non-human primates by mosquitoes [9]), and some do not require an intermediate host (e.g., Nipah virus, transmitted directly from exposure to bats [10,11]). Numerous complex interactions and predation cycles among animals produce ample opportunity for a pathogen to jump to, and cause disease in, a new host species.

Zoonotic transmission from a novel virus does not necessarily result in human disease; for example, the virus may be unable to infect human cells or evade the innate immune system [3]. However, the study of zoonotic infections in immunocompromised patients has been increasingly recognized as important because these patients may inadvertently become reservoirs for rapid viral adaptation. Examples include the accumulation of escape immune mutations during chronic SARS-CoV-2 infection of an immunocompromised cancer patient [12] and reactivation of latent herpesviruses in organ transplant recipients [13].

Patients infected with blood-borne pathogens such as HIV, hepatitis B virus (HBV), hepatitis C virus (HCV), and hepatitis delta virus (HDV) (collectively termed “HxV” herein) represent immunodeficient populations living with chronic infections. Much of the worldwide HxV burden [14,15,16] exists in areas where there is more contact between humans and wildlife [17] and thus a higher prevalence of zoonotic spillover [18]. While the primary focus of many HxV genomic surveillance programs is to identify new mutations in known viral pathogens that may impact viral infectivity, disease severity, or the sensitivity of existing diagnostic tests, these surveillance efforts can be supplemented with agnostic metagenomic next-generation sequencing (mNGS) approaches to detect other pathogens and discover novel viruses. This technology has resulted in an explosion of newly identified viral species and families over the last two decades [19,20,21,22,23,24].

In this study, we focus on two HxV genomic surveillance patients from Cameroon harboring sequence-diverse reads from nine viruses, five of which had not been previously characterized. Notably, these viruses matched more closely with those previously detected in insect, bird, and bat reservoirs.

## 2. Materials and Methods

### 2.1. Specimen Sourcing

Clinical specimens of whole blood and plasma from blood donors and participants seeking voluntary testing in Cameroon were collected in 2015–2019 with informed written consent for participation in an HxV surveillance and seroprevalence study, which was approved by the Ministry of Health of Cameroon, the Cameroon National Ethical Review Board, and the Faculty of Medicine and Biomedical Science IRB [25,26,27]. Clinical plasma specimens from the Democratic Republic of the Congo (DRC) were collected in 2017–2019 from patients seeking healthcare in the greater Kinshasa area. Informed verbal consent was obtained for participation in an HIV viral diversity study, which was approved by the Université Protestante au Congo ethics committee in Kinshasa and the University of Missouri-Kansas City Research Board [28,29]. Specimens were tested locally with HIV rapid tests before shipment to Abbott for further characterization in both studies.

Altogether, 9586 plasma specimens were screened for viremic infections with Abbott’s HIV RealTime Viral Load, ARCHITECT HIV Combo Ag/Ab test, HBV RealTime Viral Load, ARCHITECT HBsAg Qual II, HCV RealTime Viral Load, or research-use only HDV viral load qPCR assay [30], depending on available volume. Assays were performed according to the manufacturer’s instructions and reported in units of log copies per milliliter (log cp/mL) or log international units per milliliter (IU/mL) for viral load detection or signal-to-cutoff ratio (S/CO) for antigen or serological detection. Aliquots of a subset of 726 specimens were initially sent to UCSF for pathogen detection and discovery using mNGS. Remaining untouched aliquots of 22 specimens with putative divergent viral reads were later sequenced and analyzed at Abbott Laboratories, Abbott Park, IL, USA.

### 2.2. Nucleic Acid Extraction

Aliquots of plasma were initially processed at UCSF. Total nucleic acid (TNA) was extracted from 400 µL of plasma using the EZ1 Advanced XL BioRobot and EZ1 Virus Mini Kit (Qiagen, Germantown, MD, USA), and eluted in 60 µL AVE buffer. An aliquot of extracted TNA (25 µL) was treated with Turbo DNase at 37 °C for 60 min, purified with an RNA Clean & Concentrator Kit (Zymo Research, Irvine, CA, USA), and eluted in 32 µL water.

A separate, untouched aliquot of plasma was later processed at Abbott Laboratories. TNA was extracted from 500 µL of Benzonase-treated plasma (718 µL plasma + 80 µL 10× buffer + 2 µL Benzonase) with an *m*2000sp workstation (Abbott Molecular, Des Plaines, IL, USA) and eluted in 50 µL of elution buffer using a research-use only RNA/DNA protocol.

### 2.3. DNA Library Synthesis

Nucleic acids (TNA and RNA) extracted at UCSF were prepared using metagenomic sequencing with spiked primer enrichment (MSSPE) [26]. This technique combines unbiased random hexamer-primed cDNA synthesis with targeted enrichment via the spiking in a pool of HxV and arbovirus-specific primers. The random hexamers (catalog number N8080127, Thermo-Fisher Scientific, Waltham, MA, USA) and specific primers (identities available in Deng et al. [26]) were used in conjunction with the SuperScript III first-strand system (ThermoFisher Scientific, Waltham, MA, USA) for first-strand synthesis and Sequenase v2.0 polymerase (ThermoFisher Scientific, Waltham, MA, USA) for second-strand synthesis. The resulting DNA or cDNA from RNA was purified using a DNA Clean & Concentrator Kit (Zymo Research, Irvine, CA, USA) and eluted in 7.5 µL. This was followed by sequencing library construction using a Nextera XT library prep kit and custom i7 and i5 barcoding indexes (Integrated DNA Technologies, Redwood City, CA, USA). The prepared libraries were purified using AMPure XP magnetic purification beads (Beckman Coulter, Brea, CA, USA) and eluted in 20 µL of resuspension buffer (Illumina, San Diego, CA, USA).

The TNA extracted at Abbott was prepared for NGS in an unbiased fashion. First, RNA was converted into cDNA using a qScript XLT cDNA SuperMix kit (Quantabio, Beverly, MA, USA). The resulting product consisting of cDNA and extracted DNA was purified using a DNA Clean & Concentrator Kit (Zymo Research, Irvine, CA, USA) and eluted in 7 µL elution buffer. Sequencing libraries were constructed using a sparQ DNA Frag & Library prep kit (Quantabio, Beverly, MA, USA) in conjunction with *IDT for Illumina* TruSeq Unique Dual Indexes (Illumina Corp., San Diego, CA, USA). Prepared libraries were purified using AMPure XP magnetic purification beads (Beckman Coulter, Brea, CA, USA) and eluted in 20 µL of resuspension buffer (Illumina, San Diego, CA, USA).

### 2.4. Next-Generation Sequencing

NGS libraries were assessed for size and concentration using a TapeStation 2200 Bioanalyzer (Agilent Technologies, Santa Clara, CA, USA) and Qubit 2.0 Fluorometer (ThermoFisher, Waltham, MA, USA), respectively. Libraries were diluted to 1.1 nM before equimolar pooling and denaturation and final dilution to 10 pM loading concentration. Sequencing was performed on a MiSeq (at Abbott Laboratories) or HiSeq (at UCSF or Novogene) using 2 × 150 bp paired-ended chemistry.

### 2.5. Genome Assembly

FASTQ read files generated via NGS were uploaded to the SURPI bioinformatics pipeline [31], which utilizes the software tools SNAP [32], RAPSearch [33], and ABySS/Minimo [34,35] for known virus identification, divergent virus identification, and contig assembly, respectively. As a secondary analysis measure, reads that did not match any known viral references using SNAP were passed to a separate proprietary pipeline (DiVir2) for divergent virus read/contig detection using MEGABLAST [36], PSI-BLAST [37], and ABySS/Minimo [34,35].

NGS reads were imported into the CLC Genomics Workbench (Qiagen) to be mapped against contigs or reference sequences. In some cases, full genomes could be recovered from the de novo assembly carried out by ABySS/Minimo, CLC Genomics Workbench, or SPADES [38]. In other cases, iterative read mapping, contig extension, and contig joining using CLC Genomics Workbench’s Genome Finishing Module were necessary. For canonical viruses with less sequence divergence, mapping to known reference sequences was sufficient to generate full genomes. Statistical analyses of genomic coverage and annotation of open reading frames were also performed in CLC Genomics Workbench. Finalized genomes were submitted to GenBank (see Data Availability Statement).

### 2.6. Minor Variant Analysis

For each novel viral genome, a minor variant analysis was performed to determine the relative proportion of minority viral quasi-species. Briefly, all reads were first mapped back to the final consensus sequences. The “Low Frequency Variant Detection” module in CLC Genomics Workbench was used to calculate statistically significant variations in each consensus base call using the default required significance of 3% and a minimal coverage depth of 10×. The “minor variant prevalence” is calculated as the frequency of the minor variant at statistically significant sites (a value between 0 and 1) multiplied by the average coverage at the site in question.

### 2.7. Phylogenetic Analysis

Viral protein sequences were compared to the *nr* database using the BLASTp algorithm [39]. Top hits, annotated references, and outgroups were downloaded from GenBank along with relevant metadata. Domains of interest were aligned using MAFFT [40]. The evolutionary history of the aligned sequences was inferred using the maximum likelihood (ML) method implemented in IQ-TREE v2.1.3 [41]: ModelFinder [42] was used to choose the best substitution model followed by initial tree building via a stochastic algorithm and tree refinement by the nearest-neighbor interchange heuristic method [43]. The optimized ML tree was then subjected to 1000 replicates of ultrafast bootstrapping (UFB) [44] to provide statistical support for the branching topologies. Trees were appropriately rooted using Dendroscope [45] and visualized using the *ggtree* package implemented in the *R* programming language [46].

### 2.8. Putative Host Assignment

A script in the *R* programming language was written to compute mononucleotide and dinucleotide frequencies in picorna-like viral genomes and perform a discriminant analysis [47] to associate these frequencies with a particular host group. First, a mononucleotide and dinucleotide parser and frequency calculator was written using the *stringr* package (https://github.com/tidyverse/stringr/, accessed on 19 December 2021). The variance-normalized frequencies of each mononucleotide and dinucleotide were used as the predictive factors in a linear discriminate analysis (LDA) to infer the host ranges of the novel viruses [48]. The LDA was performed using the *mda* package (https://github.com/cran/mda, accessed on 1 June 2022); the training dataset utilized 945 RefSeq-quality (https://www.ncbi.nlm.nih.gov/refseq/, accessed on 30 November 2022) full-length genome sequences with annotated host from +ssRNA viruses within the phylum Pisuviricota, which contains the classes Pisoniviricetes (containing the *Nidovirales*, *Picornavirales*, *Caliciviridae*, *Solemoviridae*, *Alvernaviridae*, etc.) and Stelpaviricetes (containing the *Astroviridae* and *Potyviridae*) [24]. The dsRNA picobirnaviruses and partitiviruses were omitted from this analysis due to their different genomic structure, though their RNA replication enzyme phylogenetically clusters within the class Stelpaviricetes. The testing dataset contained new viruses with unknown hosts.

Additionally, we replicated the method of Mollentze et al. [49] to estimate the zoonotic potential of the novel viruses using a training dataset of 861 viruses with known ability or inability to infect humans (https://github.com/Nardus/zoonotic_rank/; accessed on 9 January 2023). Genome sequences of the novel viruses were collated together into a single FASTA file and coordinates of open reading frames were collected into the appropriate metadata file; these files were provided to the *PredictNovel.R* script. The numerical data output was plotted using the *ggplot2* package [50].

## 3. Results

### 3.1. Identification of Multiple Viruses in HxV-Positive Specimens

A panel of 9586 plasma specimens was collected from Cameroon and the DRC as part of a combined hepatitis virus and retrovirus surveillance study amongst blood donors and people seeking healthcare. A subset of 726 of these, consisting of HxV negative individuals and those infected with human immunodeficiency virus (HIV) 1 or 2, hepatitis C virus (HCV), and/or hepatitis B virus (HBV), was selected for screening using NGS. Extracted RNA was converted to DNA libraries and enriched using the “spiked primer enrichment” (MSSPE) approach that couples random priming with specific priming for simultaneous detection of known blood-borne virus and identification of potential novel viruses [26]. HxV viruses detected by mNGS reflected the results of virus-specific conventional serologic or molecular in vitro diagnostic (IVD) tests (Figure 1).

As expected, mNGS of HIV/HBV/HCV agreed with PCR- or antigen-based IVD tests, wherein 90.4% of IVD-positive specimens had detection of viral reads by mNGS; this could be explained by lower NGS sensitivity or some individuals being on therapy (e.g., HIV Ag/Ab combo positive but viral load negative). Approximately 44% of HxV-positive individuals (n = 288) were co-infected with some combination of HIV, HBV, or HCV. In addition to the expected blood-borne pathogens and due to the fact that the MSSPE enrichment method also includes random hexamers in its cDNA synthesis step, we detected a modest number of co-infecting viruses in this cohort, including dengue virus (n = 14), hepatitis A virus (n = 1), rhinovirus C (n = 1), parechovirus A (n = 1), influenza A virus (n = 3), and hepatitis delta virus (n = 18; exclusively in patients co-infected with HBV). The SURPI bioinformatics pipeline [31] flagged 22 specimens with NGS reads corresponding to putative novel viruses sharing low identity to known isolates.

Of these 22, specimens U172329 and U172471 from Cameroon were selected for further investigation due to the detection of divergent reads across multiple viral families. Specimen U172329 was drawn from a 30-year-old male in November 2017 who was found to be HIV-1-positive (HIV viral load negative, but HIV Ag/Ab-combo positive with an S/CO of 3.86), HBV-positive (HBV viral load of 2.97 log IU/mL, HBsAg positive with S/CO of 3704.01), and HDV positive (HDV viral load of 1.28 log IU/mL). Though negative by viral load testing, a 46%-complete genome of HIV-1 was assembled from 319 NGS reads and putatively assigned to subtype D. The full genomes of HBV and HDV were assembled through NGS and classified as genotypes E and 1, respectively, with highest identity (>96%) to strains from Cameroon. Specimen U172471 was drawn from a 29-year-old male in December 2017 who was found to be HIV-1-positive (HIV viral load negative, but HIV Ag/Ab-combo positive with an S/CO of 6.62), HBV-positive (HBV viral load of <1.0, and HBsAg positive with an S/CO of 53.83), and HDV-negative (HDV viral load test negative). A 42%-complete genome of HIV was assembled from 197 NGS reads and putatively classified as a subtype AC recombinant with highest identity to strains from Kenya and Uganda. NGS reads corresponding to HBV could not be recovered from this specimen. Both individuals were presumed healthy as they were seeking to donate blood.

Untouched aliquots of these two specimens were processed at a different location (Abbott) using a different extraction protocol (i.e., extracting both RNA and DNA), library preparation kit, and sequencing approach (fully agnostic metagenomic NGS rather than MSSPE) and produced the same divergent viral hits, corroborated the HIV/HBV findings, and additionally identified hits in DNA virus families such as *Parvoviridae* and *Circoviridae* (note that the MSSPE libraries were treated with DNase so that reads from DNA viruses would be expected to be absent from these libraries). When combining the datasets from multiple rounds of sequencing, a total of 33.2 million and 49.9 million reads were collected for specimens U172329 and U172471, respectively, from which 9 non-HxV viruses were assembled (Figure 2).

### 3.2. Detection of Known and Divergent Insect-Related Viruses

Several of the reads found in either U172329 and/or U172471 shared sequence homology with presumed insect viruses (Figure 2c,d). Predation and other animal-to-animal contact cycles are possible routes for these viruses to come into contact with humans (Figure 2b). Below, we describe these viruses in detail.

#### 3.2.1. Gemykibiviruses

Human gemykibivirus 2 is a recently-described circular virus (i.e., circovirus) of the family *Genomoviridae*. It has been isolated from different human body systems (e.g., blood, nervous, reproductive, gastrointestinal), multiple continents, and as co-infections with blood-borne pathogens such as HIV [51,52,53]. The genomes consist of two ORFs encoding a capsid protein and replication protein. Full genomes of human gemykibivirus 2 were assembled from both U172329 and U172471 at 1486× and 475.8× coverage depth, respectively (Figure 2c and Appendix A). At the nucleotide level, the two genomes are 99% identical to each other (21 total SNPs) and 98% identical to 16 strains that have been identified in humans. A second complete gemykibivirus genome was recovered through de novo assembly from U172471 at 228× average coverage depth (Figure 2c and Appendix A). The total genome is 79% identical at the nucleotide level to a circular virus isolated from a bird fecal metagenome; however, the replication protein bears ~87% nucleotide identity to the human gemykibivirus 2 replication protein.

#### 3.2.2. Flavi-like Viruses

Flaviviruses typically contain a monopartite genome, although a newly classified *Jingmenvirus* genus has tetrapartite genomes. Jingmenviruses have been identified in ticks, flies, and nematodes with a worldwide distribution [54], and some isolates may cause human disease [55]. We assembled the genome of a new *Jingmenvirus* isolate from specimen U172471, consisting of four complete segments: segment 1 (NS5) at 22×, segment 2 (VP4 and VP1) at 59.6×, segment 3 (NS3) at 45.4×, and segment 4 (VP2 and VP3) at 42× average coverage depth (Figure 2d and Appendix A). All segments share the same top BLAST hit (on average, 83% amino acid identity and 77% nucleotide identity) to the Shuangao insect virus 7, isolated from a pool of flying insects from eastern China [54].

#### 3.2.3. Densoviruses

The densoviruses are a subfamily within the single-stranded DNA viral family Parvoviridae. They are known to infect arthropods (including insects and crustaceans) and echinoderms. They have also been identified in the fecal virome of mammals such as rodents [56]. We assembled full genomes of a novel densovirus from both U172329 and U172471 with average coverage depth of 176.1× and 28.8×, respectively (Figure 2c and Appendix A). The two sequences are 99.99% identical and share similar coverage depth profiles, with only 5 SNPs detected across the full genome length of 6520 bp. The genomes each contain four open reading frames (ORFs) on the sense strand in the following order: NS2, putative NS3, NS1, VP1. A BLASTp comparison of the amino acid sequence of NS2 reveals 50% identity and 66% similarity to the closest known relative, a densovirus recently isolated from the fecal virome of birds in China. Other top BLASTp hits come from viruses isolated from spider silk glands (e.g., false wolf spider monodnaparvovirus) and rodent feces (e.g., Fresh Meadows densovirus 1). On average, the four ORFs bear 38% amino acid identity/56% nucleotide identity to their single closest relatives.

#### 3.2.4. Nodaviruses

Nodaviruses are bisegmented positive-sense RNA viruses. They have been identified in both invertebrates and vertebrates and have been linked with disease in insects (genus *Alphanodavirus*), fish (genus *Betanodavirus*), and crustaceans (genus *Gammanodavirus*) [57]. Two distinct nodaviruses were assembled from U172471. The first virus is most closely related to the Shuangao insect virus 11, detected in an insect pool isolated in China in 2013 [23]. Only the first segment encoding Protein A (the replicase) was recovered at 77.1× average coverage depth (Appendix A). Protein A bears 67% amino acid identity and 42% amino acid identity to its homologs from Shuangao insect virus 11 and the Flock House virus, respectively (Figure 2d).

The second nodavirus is nearly identical to Porcine nodavirus strain IA/2017 (genus *Alphanodavirus*), isolated from the brain tissue of pigs with neurologic signs. Segment 1 was recovered at 89% coverage length/6.6× coverage depth and segment 2 was recovered at 89% coverage length/8.2× coverage depth. The protein A ORF bears 99% amino acid identity/99% nucleotide identity, and the capsid precursor ORF bears 100% amino acid identity/99% nucleotide identity to strain IA/2017 (Figure 2d and Appendix A).

#### 3.2.5. Picornaviruses

Multiple divergent picornavirus contigs were also detected for which only partial genomes were assembled. Coverage for various genes or domains (e.g., capsid, RdRp, helicase, etc.) shared weak identity to spider or bat viruses, including Washington bat picornavirus (GenBank accession NC_030843), and Burke-Gilman virus (GenBank accession NC_031693.1) (Figure 2d). Despite the relatedness of these two viruses and with contigs spanning anywhere from 500 to 4000 amino acids, they were not able to be assembled into a single discrete genome (Appendix A). These highly divergent viruses bear only ~40% amino acid identity to their closest relatives.

### 3.3. Detection of Viruses with Potential for Vertebrate Infection

Beyond the viruses described above, U172329 and/or U172471 contained 3 additional viruses with potential for vertebrate infection: a dicistrovirus, a cyclovirus, and a bastrovirus (Figure 3).

#### 3.3.1. Dicistroviruses

The dicistroviruses are a sister family to the picornaviruses (order Picornavirales). Invertebrates generally serve as natural hosts, although there are studies describing human febrile illness putatively associated with dicistroviruses [58,59]. A full genome of a dicistrovirus was assembled from specimen U172471 at 11.8× average coverage depth. The genome has a length of 9965 bp and contains two ORFs and the canonical dicistrovirus internal ribosome binding site (IRES) elements (Figure 3a, first and second rows). A second dicistrovirus was assembled from specimen U172329 at 45% total coverage and 1.1× average coverage depth (Figure 3a, first and second rows). ORF1 encodes a non-structural polyprotein (1793 AA, ~205 kDa) bearing 98% amino acid identity and 94% nucleotide identity to a dicistrovirus associated with a febrile patient from Peru (Genbank accession AWK23470.1) [59] and ORF2 encodes a structural polyprotein (807 AA, ~90 kDa) bearing 99% amino acid identity/96% nucleotide identity to the same reference. Genomic analysis revealed only a single minor variant site within the read mapping (Figure 3a, third row). The two isolates appear to be 88% identical to each other at the nucleotide level across the 45% of the genomes that could be aligned (Figure 3a, fourth row), although the low genome coverage from U172329 places uncertainty over this value. Maximum likelihood phylogenetic reconstruction of the complete ORF1 from U172471 reinforces its close association to the human blood-associated dicistrovirus (Genbank accession AWK23470.1) isolated from febrile patients in Peru (Appendix A). These viruses appear to be unrelated to dicistroviruses previously detected in febrile pediatric patients in Tanzania [58].

#### 3.3.2. Cycloviruses

Viruses in the family *Circoviridae* are non-enveloped, circular, single-stranded DNA viruses that have been found in insects, birds, and mammals, including humans [60]. Two complete and identical genomes of a new cyclovirus were assembled from U172329 and U172471 at 176.1× and 28.8× average coverage depth, respectively (Figure 3b, first and second rows). The genomes consist of a single 1783-bp circular segment containing two ORFs encoding a DNA polymerase/replicase (*rep*) and a capsid protein (*cap*), as well as a replication initiation site. A genomic analysis indicated several minor variant sites reside within capsid and the untranslated region (Figure 3b, third row). These two genes share the same top BLAST hit, Mongoose-associated cyclovirus strain Mon-20 (Genbank accession MZ382573), isolated in 2017 from the feces of an Indian mongoose (*Urva auropunctata*) in St. Kitts and Nevis [61]. The *cap* protein (222 AA, ~25 kDa) bears 86% amino acid identity and 86% nucleotide identity to this reference, while the *rep* protein (277 AA, ~32 kDa) bears 100% amino acid and nucleotide identity to this reference. The second top BLAST hit is Cyclovirus isolate CyV-LysokaP4/CMR/2014 (Genbank accession MG693174), isolated in December 2013 from the feces of a Straw-colored fruit bat (*Eidolon helvum*) in Cameroon. A maximum likelihood reconstruction placing these new cyclovirus isolates within the family *Circoviridae* and genus *Cyclovirus* shows that they belong to a well-supported clade containing other cycloviruses found in bat gastrointestinal and respiratory tracts, as well as rodent, human, and bird gastrointestinal tracts (Appendix A). The Cyclovirus-VN strains which have been previously isolated from human cerebrospinal fluid and plasma [62] belong to a different clade than the virus described here.

#### 3.3.3. Bastroviruses

Bastroviruses (“Basal astrovirus”) comprise a recently described group of single-stranded, positive-sense RNA viruses related to astroviruses and the hepatitis E virus [63]. Bastroviruses and their relatives have been identified in raw sewage and the feces of mammals, birds, and invertebrates, sometimes being observed in association with both asymptomatic and symptomatic gastrointestinal infection [23]. A full genome of a bastrovirus was assembled from specimen U172329 at a coverage depth of 71.7×. A second genome with 97% nucleotide identity to the first was assembled from U172471 at 87% coverage length and 9.4× coverage depth. The complete genome is a single 5968-bp segment containing 3 ORFs. Genomic analysis indicates a significant number of minor variant sites in these genomes, with most appearing in the C-terminus of ORF1 and the N-terminus of ORF2. The three ORFs share the same closest relative, a bastrovirus isolated in 2018 from shellfish in Cameroon (Genbank accession MW924353). ORF1 encodes a non-structural polyprotein (1,407 AA, ~159 kDa) containing methyltransferase, helicase, and RdRp domains, and bears 97% amino acid identity and 90% nucleotide identity to reference MW924353. ORF2 encodes a structural polyprotein (350 AA, ~37 kDa) and bears 96% amino acid identity and 90% nucleotide identity to reference MW924353. The putative ORF3 (113 AA, ~13 kDA) contains a domain of unknown function and bears 87% amino acid identity and 91% nucleotide identity to an unannotated, but putative, ORF3 from reference MW924353.

Due to the relatedness between bastroviruses and human disease-causing astroviruses (*Astroviridae*) and hepatitis E-like viruses (*Hepeviridae*), as well as the relative lack of phylogenetic data for bastroviruses, we reconstructed an ML phylogeny of these three groups (Figure 4). All available amino acid sequences of the RdRp domain of ORF1 from viruses annotated as “bastrovirus” were compared to representative sequences from the astroviruses and hepatitis E-like viruses (Figure 4a). Data regarding sampled host, isolation source, genome organization, and capsid type (deduced from the GenBank metadata, BLASTp searches, and *pfam* protein family prediction [64]) were also integrated to provide comparative analysis. Isolates currently annotated as “bastrovirus” were observed to form a paraphyletic group branching closer to the *Hepeviridae* than to the *Astroviridae*. Inspection of the clades of bastroviruses reveals similarities in genome organization, but differences in capsid relationships. The first bastroviruses discovered, Bastroviruses 1-7 from human stool [63], and close relatives form a monophyletic group (labeled “Bastroviruses” in Figure 4a); members of this group contain capsids with a high level of similarity to those from the *Astroviridae*. Viral genomes in this monophyletic group have been exclusively recovered from the gut virome of vertebrates.

Another clade with strains annotated as “bastrovirus” (labeled “Unclassified Bastroviruses” in Figure 4a) contains members recovered from vertebrate guts as well as from invertebrates, aquatic sediments, and raw sewage. Within this clade is a monophyletic group containing the *Alphatetraviridae*, which are known insect-vectored viruses with a different capsid type and genomic organization than the other viruses presented in the tree. The “unclassified bastroviruses” seem to contain a capsid type distinct from the *Astroviridae* and *Herpeviridae*, bearing loose resemblance to calicivirus capsid proteins. The sub-group containing the bastroviruses from U172329 and U172471 (denoted by a star in Figure 4a and expanded in Figure 4b) has a worldwide distribution; however, the closest relatives have been found in mollusks from Cameroon, raw sewage in Brazil, mosquitos from northern California, and bat feces from Vietnam. In this closely related group, aside from the virus from U172329 and U1712471, any viruses isolated from vertebrates are from the gut virome of insectivorous bats and birds (the viruses from U172329 and U172471 are the first in this clade to be identified in humans or in vertebrate blood). Based on these patterns, it is probable that most of the members of this clade have invertebrate hosts and can thus be found in the gut virome of vertebrates that consume those invertebrates.

### 3.4. Evaluation of Possible Host Range

Attempts to culture viruses directly from leftover aliquots of patient plasma on human and bat cell lines were unsuccessful. An alternate in silico evaluation was performed as described elsewhere [47,48]. First, we considered the overall mono- and dinucleotide content [65] as proxies for host range using linear discriminant analysis (LDA). As positive single-stranded RNA (+ssRNA) picorna-like viruses have been analyzed in this way in the past, we restricted this analysis to the bastroviruses and dicistroviruses. We compared the mono- and dinucleotide content of these viral genomes against a training dataset containing 945 well-annotated +ssRNA reference sequences with confirmed host range (restricted to vertebrates, invertebrates, and plants; Appendix A) from the phylum *Pisuviricota*, which contains (but is not limited to) such groups as nidoviruses, coronaviruses, picornaviruses, caliciviruses, astroviruses, and potyviruses. The LDA incorporated all 4 mononucleotide frequencies and all 16 dinucleotide frequencies to determine the most suitable contribution of each for achieving host classification. The results, graphed in a canonical score plot (Figure 5a), show a reasonable separation of the 90% confidence ellipses for each host class.

When the bastroviruses from U172329 and U172471 and the dicistrovirus from U172471 are compared against the training dataset, they fall within the 90% confidence ellipsis for the invertebrate host class, inferring that the natural host for these viruses are indeed invertebrates. We also highlight other viruses with phylogenetically close relationships to the bastroviruses from U172329 and U172471 for the sake of comparison. Hepatitis E virus and mamastrovirus 1 clearly cluster with vertebrate viruses, as expected. Bastrovirus Brazil/sewage, a close relative to the Cameroonian strains obtained from an environmental sample, clusters within the invertebrate viruses. On the other hand, bastrovirus VietNam/Bat/17918/21, another close relative isolated from bat feces, falls just within the 90% confidence ellipsis for the vertebrate host class. This implies that this strain, though belonging to a monophyletic group with a presumed insect host, may be in the process of adapting to a new vertebrate host.

A previous study from Mollentze et al. [49] established a methodology for estimating the human disease potential for viruses, irrespective of family, based upon multiple classes of information, including mono- or dinucleotide content, phylogenetic/taxonomic relationships, similarity to human housekeeping transcripts, and similarity to interferon-stimulated genes. Application of these feature sets together allowed for calculation of a single mean value representing zoonotic probability. We applied this algorithm to predict the zoonotic probability for all the viruses identified from specimens U172329 and U172471, and other well-studied viruses found in the same families for comparison. We applied the mean probability cutoff of 0.293 as previously suggested [49] to balance sensitivity and specificity. Among the viruses found in U172329 and U172471, only the novel cyclovirus and porcine nodavirus scored with a mean zoonotic probability over the cutoff (i.e., a priority class of “high” or “very high”), though all others except for the gemykibiviruses had upper 95% interquartile ranges that surpassed the cutoff. Encouragingly, the model predicted mean zoonotic potentials over the cutoff for known human pathogens dengue virus, mamastrovirus 1, hepatitis E virus, human cyclovirus VS5700009, parvovirus B19, and likely human pathogen Jingmen tick virus. However, the Flock House virus (Nodaviridae), Israel acute paralysis virus (Dicistroviridae), and Cricket paralysis virus (Dicistroviridae) also scored a mean zoonotic probability over the cutoff, even though these viruses have never been documented to infect humans. Taken together, these in silico analyses suggest that despite recovery from human plasma, most of the new viruses have sequence signatures denoting a low potential for infecting humans.

## 4. Discussion

Metagenomic next-generation sequencing has revolutionized the speed and scale at which new viruses are discovered [22,23,66]. The process of assembling genomes that once took years to complete, involving isolation in culture and sequencing from PCR amplicons in gels, has been distilled down to days or even hours [22], often without the need to pick up a pipette [67]. As the push to adopt mNGS for diagnosis of infections in patients gains momentum, there are significant practical considerations that stand in the way of its immediate realization [68,69,70], including but not limited to sensitivity, the validation of a limitless number of targets, adoption of standardized workflows and reference materials, and cost. Our paper highlights what is both a strength and weakness of metagenomics: the means to sequence any microbe in a patient specimen. While we detected multiple viruses, it was not possible to retrospectively ascribe any of these to a disease, let alone inform patient management [69]. Indeed, numerous studies report the presence of a growing list of viruses (e.g., torque teno virus, pegivirus) in metagenomic libraries from individuals with or without symptoms. Without clinical context and confirmatory orthogonal testing (e.g., PCR, serology, culture), it is challenging to equate mNGS detection of a new virus with identification of a bona fide human pathogen, as opposed to a contaminant, commensal partner, or colonizer [20].

We believe that the viral findings in specimens U172329 and U172471 are unlikely to be due to inter-specimen or laboratory contamination. While the two individuals are connected by virtue of being sampled at the same clinic in Cameroon within a one-month period, they have distinct HIV/HBV/HDV diagnostic/genotypic profiles and distinct viral reads. For the +ssRNA viral families shared between the specimens, the assembled sequences are similar, but not identical. Additionally, multiple RNA families are found in U172471 but not U172329 (e.g., Nodavirus, Jingmenvirus). While the assembled sequences for the DNA viral families shared between the specimens (i.e., Gemykibivirus, Cyclovirus, Densovirus) are 99–100% identical, the comparatively slower rate of DNA virus evolution [71] offers a potential explanation. Contamination during NGS library preparation is deemed unlikely due to the reproducibility of the obtained viral reads across separate aliquots, separate laboratories, separate library preparation techniques, and separate sequencing instruments. A second specimen (U172436) collected at the same clinic and on the same day as U172329 was also analyzed by mNGS and did not contain any of the divergent viral reads described in this study. We acknowledge, however, that these factors do not rule out sporadic exogenous contamination of patient plasma at the time of blood draw or of the collection tubes or pipettes during specimen handling or processing.

Clinical data is not available for the individuals of interest beyond the results from HxV IVD testing. Thus, we cannot know why they presented to the clinic or if they were sick at the time of blood draw or in the following days. We also do not know their immunocompromised status; while the patients were positive for HIV by Ag/Ab testing, HIV viral load was negative, suggesting that the individuals may have been on antiretroviral therapy. Thus, in the absence of further clinical data, we cannot assume that any of the viruses found were causing illness in the patients. If some of these viruses are infectious to humans, potential routes of exposure include (1) bites from infected invertebrates, (2) consumption of food or water contaminated with invertebrates, or (3) consumption of food or water contaminated with excrement from birds, bats, or rodents who feed on invertebrates. Several reports have presented recovery of similar groupings of viral families from bird, bat, and rodent excrement [56,72,73,74,75,76,77,78,79,80]. However, in the current study, we detected sufficient genetic material from these viruses in human plasma to assemble full genomes at high coverage. Furthermore, the presence of minor variants in the genomes, especially in the bastroviruses and cycloviruses, suggest recent viral replication and quasispecies formation as a response to immune pressure, either in the true host (invertebrates) or accidental host (humans) [81].

Some of the detected viruses in this study derive from viral families that may possibly confer an increased risk for zoonosis, including the bastroviruses, dicistroviruses, and cycloviruses:**Bastrovirus**: In describing the initial discovery of bastroviruses in human stool, Oude Munnink et al. [63] suggested that sustained PCR detection over decades and accumulated genetic diversity in the capsid proteins showed the viruses had been circulating in humans or another host for some time. Bastroviruses are also prevalent globally, with recent detection in North America [82], South America [83], Asia [84], Oceania [85], and Africa [72]. Notably, the most closely related isolates to our presented bastroviruses were found in Cameroonian shellfish [86], sampled within 200 km of where our human subjects were located. As the genomic sequences are 97% identical at the amino acid level, it is possible that these viruses have been cryptically circulating in the shellfish reservoir or human population. We observed a similar profile of viral families (e.g., cyclovirus, densovirus, picornavirus) to that observed in that same metagenomic survey [86], but also in North American shellfish [87]. These lines of evidence point to shellfish consumption being a possible source for human infections from several of the detected viruses (if these viruses are indeed infectious). Of note, no clinical disease symptoms have been statistically associated with the presence of bastrovirus [63]. However, the relatedness of bastroviruses to established human pathogens such as astroviruses and hepatitis E virus, both of which are transmitted by contaminated food or water, warrants further attention. Based on the phylogenetic analysis presented in Figure 4, the acquisition of new capsid types and new ORF3s are the likely drivers of host-jumping events (indeed, most minor variants detected in our bastroviruses appeared in the capsid, see Figure 3c). This has likely happened before with the hepatitis E-like viruses, with an ancestral invertebrate- or bird-infecting virus jumping to rodents and a later descendent jumping to bats and primates (Figure 4). More work is needed to re-classify bastroviruses and better explore the true host ranges of the various clades, especially since some members seem to be adapting vertebrate-like genomic nucleotide compositions (Figure 5).**Dicistrovirus:** While dicistroviruses are believed to exclusively infect invertebrates, they have been detected in patients with febrile illness in Peru and Tanzania [58,59]. Two different dicistroviruses were detected in U172329 and U172471 with >90% nucleotide identity to the virus found in the Peruvian patient population. Only one position in the genome from U172471 had a minor variant, suggesting that these viruses may either be contaminants or may not be replicating in the primary (e.g., insect) or incidental (e.g., human) host. Nonetheless, the zoonotic potential analysis in Figure 5 suggests potential human infectivity in other dicistroviruses, so further investigation of this virus family is warranted.**Cyclovirus:** Both U172329 and U17471 possessed an identical cyclovirus genome with most minor variants detected in capsid or the intergenic, untranslated region. This may indicate immune evasion and lack of selective pressure in the natural host, respectively. Our sequence’s closest relatives have been found in mongoose feces [61], human feces [88], human respiratory tracts [89], rodents feces [78], bat feces [72], winged insects [90], and chicken muscle [91]. Moderate identity is seen with cycloviruses isolated from human plasma, respiratory tract, and CSF, with and without associated clinical manifestations such as encephalitis, respiratory illness, and sepsis [53,62,92,93,94]. Since disparate groups of cycloviruses continue to be discovered in human specimens, we share some concern that this viral family may contain members capable of zoonotic disease. Indeed, it has been suggested that dietary and environmental sources of exposure lead to unexpected new ecological niches for small DNA viruses such as cycloviruses [95].

Carefully designed and controlled experiments replicated across independent research sites, combined with judicious skepticism and efforts to rule out contamination, are necessary to yield and interpret reliable virus discovery results. We were unable to investigate any of Koch’s postulates due to the lack of clinical data and inability to successfully culture any of the viruses in multiple cell lines using leftover plasma. We reiterate that just because a virus is found in human plasma does not necessarily mean that it is infectious or causes disease; however, it may provide useful information about the spectrum of viruses encountered by humans in regions prone to zoonotic spillovers. In the modern ‘sequence-first’ era of virology, there is a high burden of proof for demonstrating that a newly discovered agent is causative of disease: extraordinary claims require extraordinary evidence. To achieve this, ‘virus hunters’ should focus on sick populations to more readily determine whether a newly detected virus is pathogenic [96]. Phylogenetic analyses and follow-up seroepidemiologic investigation of exposed populations may also generate useful confirmatory data.

## Figures and Tables

**Figure 1 viruses-15-01022-f001:**
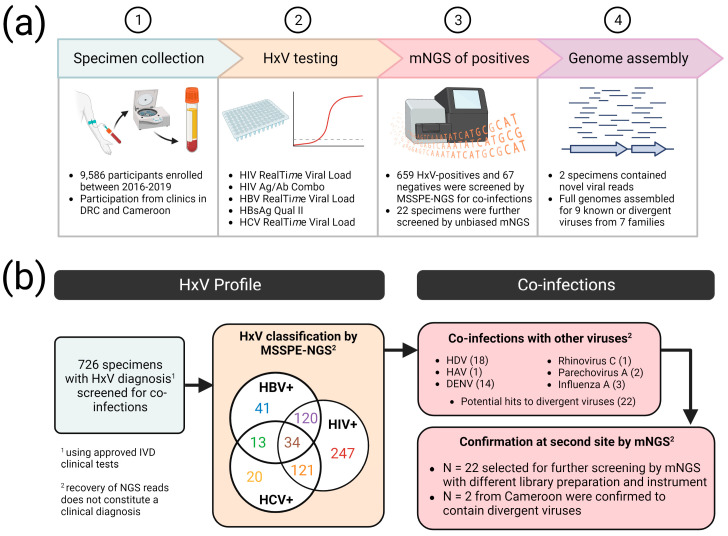
Study flow chart. (**a**) Testing regime for specimens, and (**b**) quantification of viral diagnoses and viral identifications via deep sequencing. (Figure created with BioRender.com).

**Figure 2 viruses-15-01022-f002:**
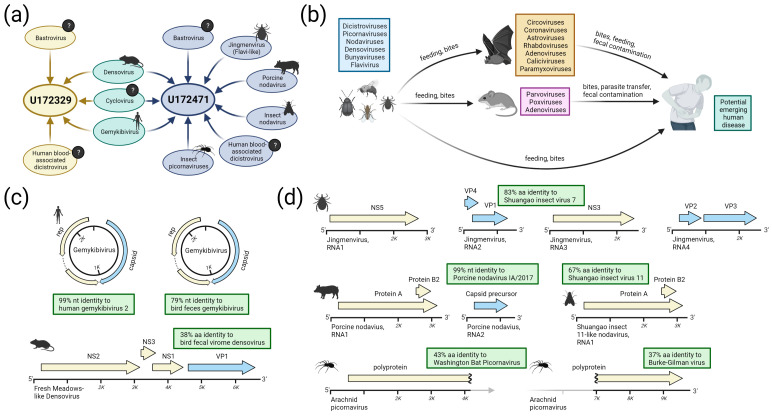
Summary of non-HxV viruses recovered from specimens U172329 and U172471. (**a**) Viral families from which a partial or complete genome was recovered from each specimen. Identical sequences of the novel densovirus, novel cyclovirus, and known human gemykibivirus 2 were found in both specimens. (**b**) Schematic depicting predation/encounter scenarios that may transmit viral genetic material from lower animals to higher animals. (**c**) Assembled genomes for known and novel DNA viruses described in panel A. (**d**) Assembled genomes for known and novel RNA viruses described in panel A. (Figure created with BioRender.com).

**Figure 3 viruses-15-01022-f003:**
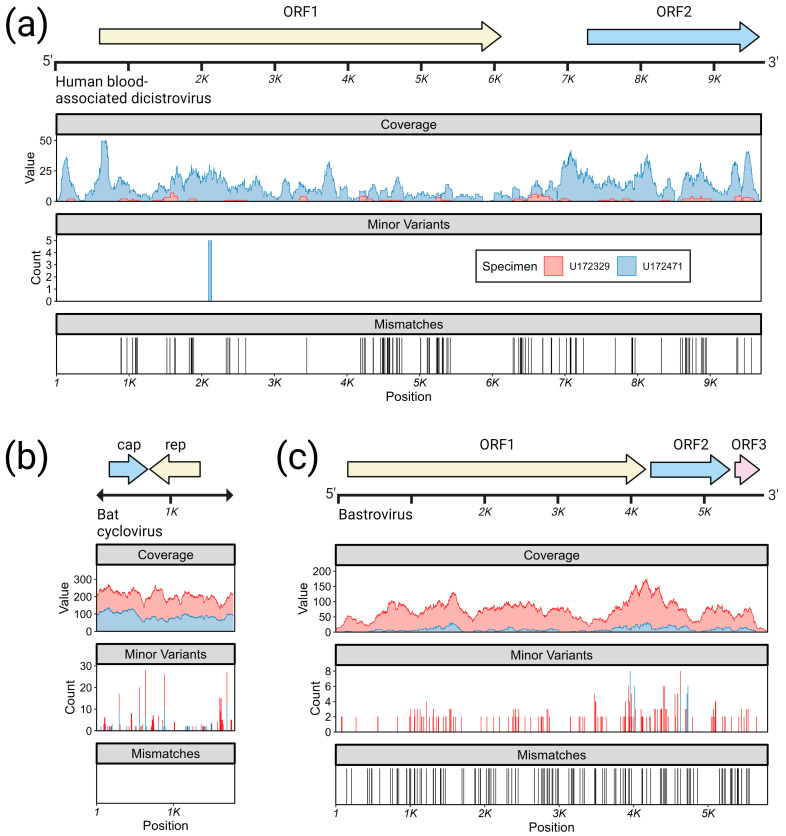
Genomic maps and mapping statistics for three viruses found in specimens U172329 and U172471 that have previously been detected in mammals: (**a**) human blood-associated dicistrovirus, (**b**) bat cyclovirus, and (**c**) bastrovirus. In each panel, mismatches represent single-nucleotide polymorphisms detected between the U172329 and U172471 isolates (Figure created using assets from BioRender.com).

**Figure 4 viruses-15-01022-f004:**
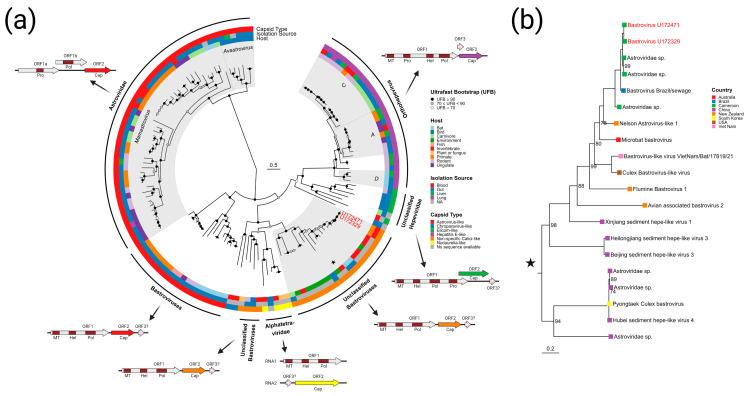
Phylogenetic reconstruction of the *Astroviridae, Hepeviridae*, and bastroviruses. (**a**) Amino acid ML phylogeny of the RdRp domain from 133 viral isolates. The amino acid sequences were aligned using the L-INS-i algorithm of MAFFT and the *Q.pfam + F + R6* substitution model was selected by IQ-TREE 2 as the most appropriate model to reconstruct the phylogeny. Metadata including sampled host, isolation source, and capsid type are shown as rings outside of the tree. Each clade is presented with a representative genomic schematic with domains of interest from each ORF indicated (domain abbreviations: MT—viral methyltransferase; Hel—helicase; Pol—RNA-dependent RNA polymerase; Pro—serine protease; Cap—capsid). The clade of interest containing the viruses isolated in this study is denoted by a star. (**b**) An expanded view of the monophyletic group denoted by a star in panel A. Taxa are labeled with isolate name and sampling country. Ultrafast bootstrap support is reported at the nodes.

**Figure 5 viruses-15-01022-f005:**
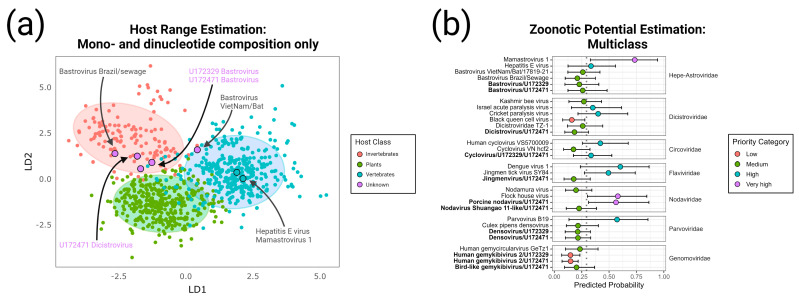
Assignment of host class and zoonotic potential. (**a**) The canonical score plot of a linear discriminant analysis used to classify picorna-like viral sequences into three host groups using all 4 mononucleotide and all 16 dinucleotide frequencies. The plot shows the separation of host groups through the two most statistically significant factors. The training dataset with known host range (n = 945 genomes) was used to establish a scoring profile such that the viral sequences with unknown host could be classified. The ellipses represent the 90% confidence level (i.e., 90% of sequences fitting a host range group fit inside the ellipsis) centered on the centroid of each group. Sequences from the bastroviruses from U172329 and U172471, the dicistrovirus from U172471, and four comparator sequences are labeled separately. (**b**) Predicted probability of human infectability for all novel viruses identified in this study and closely related comparator sequences. Dots show the mean and bars show the 95% interquartile range of predicted probabilities across the best-performing 10% of iterations. The cut-off for zoonotic potential was set at 0.293 with priority categories assigned as previously described [49]: low: mean and upper/lower interquartile ranges below cutoff; medium: mean below cutoff but upper interquartile range above cutoff; high: mean above cutoff but lower interquartile range below cutoff; very high—mean and upper/lower interquartile ranges above cutoff. In both panels, the dicistrovirus recovered from specimen U172329 was not analyzed due to its low (45%) total coverage.

## Data Availability

The genomes generated in this study have been deposited to GenBank and are available at accessions OQ835729 through OQ835746. The data and R scripts required to reproduce Figure 3, Figure 4 and Figure 5 will be provided upon request.

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
