# Peer review of "Metagenomic Detection of Divergent Insect- and Bat-Associated Viruses in Plasma from Two African Individuals Enrolled in Blood-Borne Surveillance"

_viruses, 2023, doi:10.3390/v15041022_

Round 1

Reviewer 1 Report

The manuscript entitled "Metagenomic detection of divergent insect- and bat-associated viruses in plasma from two African individuals enrolled in blood-borne surveillance" was evaluated. Researchers reported that 727 patient specimens were analyzed by mNGS to identify viral co-infections. Then divergent sequences from nine poorly characterized or previously uncharacterized viruses were also identified in two individuals. These viruses were assigned to the following groups by genomic and phylogenetic analyses: densovirus, nodavirus, jingmenvirus, bastrovirus, dicistrovirus, picornavirus, and cyclovirus. This manuscript helps to understand of potential candidates of zoonotic viruses. The paper is in the scope of the journal and may be published.

Minor comments:

Some pages were missing in references, such as references 28, 48, 56, 83, 94…and some page number styles were different.

Author Response

Thank you for your favorable response and for catching the errors with some references. We have corrected those references where appropriate; some of these are “online-only” articles and do not have formal page numbers, but rather have just an article number.

Reviewer 2 Report

In the paper entitled “Metagenomic detection of divergent insect- and bat-associated viruses in plasma from two African individuals enrolled in blood-borne surveillance”, Gregory S. Orf et al analyzed 659 plasma samples collected from HIV- and/or hepatitis-infected people in Cameroon and DRC. Assuming that immunocompromised patients living in areas where contacts between humans and wildlife are frequent may become reservoirs for rapid viral adaptation, they underwent metagenomic NGS to detect any co-infection with other pathogens. A subset of 22 samples with NGS reads matching to putative novel viruses were selected for agnostic metagenomic NGS analyses. The paper is describing the sequencing and phylogenetic analyses for nine unknown or badly known viruses that were identified in two individuals from Cameroon. They could assemble complete genomes of human gemykibivirus 2, jingmenvirus, densovirus, nodaviruses, and recovered contigs of picornavirus, which are insect-related viruses. Importantly, they also assembled dicistrovirus, a new cyclovirus and bastrovirus genomes of viruses with potential for vertebrate infection.

The study uses proven methods for the metagenomic sequencing and bioinformatic analyses, as well as for phylogenetic reconstructions. The paper is well written and well explained in all its sections, the figures and diagrams are clear and suitable, and the discussion highlights the limits of the study, the main being that ‘a virus found in human plasma does not necessarily mean that it is infectious or causes disease’. That’s why the idea that the search around zoonotic spillover should target exposed but also immunocompromised people is very interesting and promising.

I have only one question to which I would like a discussion. The authors first processed with MSSPE sequencing to enrich for the HxVs and arboviruses, and next they underwent agnostic metagenomic NGS on a subset of samples which displayed reads not related to the MSSPE targets. Don’t you think you can miss something in the non-selected samples? Because I suppose that you also got reads of HxV (HIV, hepatitis virus) at the same time as the gemykibirus or bastrovirus or other viruses with the second method, although you don’t have specified this point. Logically, the MSSPE sequencing should promote the viruses targeted by the primers (HxVs and arboVs) to the detriment of the other viruses that you are searching for. Consequently, these last viruses may be masked, hidden, for the majority of samples, even if you could see them in 22 samples.

Author Response

Thank you for your favorable response. Regarding the MSSPE-enriched sequencing approach: we would like to clarify that the MSSPE enrichment (which occurs during cDNA synthesis) includes not only primers targeting HxVs and arboviruses, but random hexamers as well – which are always utilized in a “pure” unbiased metagenomic approach. Thus, while one expects to see enrichment of HxVs and arboviruses, this should not prevent other viruses from being detected. While it is certainly true that the recovered number of HxV and arbovirus reads (in terms of specific reads per million total reads, or “RPM”) will be higher than other viruses, and non-HxV and non-arboviral reads will be relatively depleted, we can indeed still see evidence of other viruses in these specimens. So, when we reflex back to unbiased metagenomic NGS, it is primarily aimed at raising the abundance of non-HxV and non-arboviral reads and increasing the coverage of the corresponding genomes.

Another important point of this workflow is that the MSSPE-based enrichment is performed on extracted RNA, rather than TNA (total nucleic acid). Thus, DNA viruses will be missed, unless they are producing a large amount of RNA transcript, such as if they are actively infecting cells. So, reflexing back to unbiased metagenomic NGS which uses TNA as the starting material will add DNA viruses to the possible detectable species. So, in this way, the reviewer is indeed correct that some viruses – namely DNA viruses – will be missed in MSSPE-based enrichment (please see lines 247-253). To further address the concerns of the reviewer, we have added a statement on lines 223-224 of the main text.

Reviewer 3 Report

The manuscript titled “Metagenomic detection of divergent insect- and bat-associated 2 viruses in plasma from two African individuals enrolled in 3 blood-borne surveillance,” authored by Orf et al. describes the diversity of viruses in human blood samples collected in the Democratic Republic of Congo and Cameroon between 2015-2019 under a surveillance study.

The authors provided adequate and comprehensive details of each section of the manuscript. The introduction provides sufficient background information relevant to the study conducted. The methodology is comprehensively described and appears to be reproducible. The authors used a two-step approach for virus genome assembly and identification; they used the SURPI bioinformatics pipeline to identify known viruses and a separate proprietary pipeline, DiVir2, for identifying divergent virus reads/contigs that did not match the known viruses.

The authors performed minor variant and phylogenetic analyses of the virus genome sequences. Interestingly, they also estimated the zoonotic potential of the novel viruses identified in this study using a previously reported method.

The results and discussion sections comprehensively explain the diversity of viruses detected and identified in the study. The figures drawn are quite informative and grab the attention of the reader. The study adequately addresses and concludes the research questions of the study. Overall, the study is very well conducted with an appropriate research design. The manuscript is well written.

I have only a few minor suggestions for the authors to consider:

1.      Lines 57-58: “… pathogens such as human immunodeficiency virus, hepatitis B virus, hepatitis C virus, and hepatitis delta virus…”. The names of viruses may be abbreviated here at the first use rather than later in the manuscript.

2.      Lines 111-112: “The random and specific primers were used in conjunction with the SuperScript III first-strand system for first-strand synthesis and Sequenase v2.0 polymerase for second-strand synthesis.” In the methods section, it would be nice to provide the details of the random- and specific- primers used in the study.

Author Response

Thank you for your favorable response. We have addressed the abbreviations on lines 57-58. Additionally, we have added more detail to lines 109-115 regarding the primers. The specific primer identities for HxV and arbovirus enrichment can be found in the Deng et al. study (reference 26) and the random hexamers are off-the-shelf, from ThermoFisher (catalog number is now included).